# In Vitro Antiprotozoal Activity of *Schinus molle* Extract, Partitions, and Fractions against *Trypanosoma cruzi*

**DOI:** 10.3390/plants13162177

**Published:** 2024-08-06

**Authors:** Nancy E. Rodríguez-Garza, Ramiro Quintanilla-Licea, Ricardo Gomez-Flores, Lucio Galaviz-Silva, Zinnia J. Molina-Garza

**Affiliations:** 1Laboratorio de Patología Molecular y Experimental, Facultad de Ciencias Biológicas, Universidad Autónoma de Nuevo León, Ave. Universidad S/N, Cd. Universitaria, San Nicolás de los Garza 66455, Nuevo León, Mexico; nancy.rodriguezgrz@uanl.edu.mx; 2Laboratorio de Fitoquímica, Facultad de Ciencias Biológicas, Universidad Autónoma de Nuevo León, Ave. Universidad S/N, Cd. Universitaria, San Nicolás de los Garza 66455, Nuevo León, Mexico; rquintanilla.uanl@gmail.com; 3Facultad de Ciencias Biológicas, Universidad Autónoma de Nuevo León, Ave. Universidad S/N, Cd. Universitaria, San Nicolás de los Garza 66455, Nuevo León, Mexico; rgomez60@hotmail.com

**Keywords:** *Trypanosoma cruzi*, Chagas disease, *Schinus molle*, antiparasitic activity, medicinal plants, public health

## Abstract

Chagas disease, caused by the protozoan *Trypanosoma cruzi*, represents an important and worldwide public health issue, particularly in Latin America. Limitations of conventional treatment with benznidazole and nifurtimox underscore the urgent need for new therapeutic strategies for this disease. *Schinus molle*, a tree used in traditional medicine for various ailments, has demonstrated promising antiparasitic activity. The in vitro anti-*T. cruzi* activity of *Schinus molle* crude methanol extract, partitions, and fractions, as well as their cytotoxicity in Vero cells and *Artemia salina*, and hemolytic activity in human erythrocytes were assessed. Most of the extracts possessed anti-*T. cruzi* effects, with Sm-CF3 being the fraction with the highest activity (IC_50_ = 19 µg/mL; SI = 6.8). Gas chromatography–mass spectrometry analysis identified 20 compounds, with fatty acyls comprising the predominant chemical class (55%). We also identified the antiparasitic compounds cis-5,8,11,14,17-eicosapentaenoic acid and trans-Z-α-bisabolene epoxide, suggesting their potential contribution to the observed anti-*T. cruzi* activity. In conclusion, our findings support the therapeutic potential of *S. molle* as a source of novel antiparasitic agents against *T. cruzi*.

## 1. Introduction

Chagas disease, also known as American trypanosomiasis, is a parasitic zoonotic disease caused by the hemoflagellate protozoan *Trypanosoma cruzi* (Chagas, 1909) [1]. It is the most important parasitic disease in Latin America and is classified among the 20 Neglected Tropical Diseases (NTDs), with an estimated 6 to 8 million people infected worldwide, mostly in Latin America [2,3]. However, the distribution of the disease is changing due to the relocation of individuals from endemic countries. It is also estimated that more than 12,000 people die each year from this disease, and more than 75 million individuals are at risk of contracting it [2]. In Latin America, *T. cruzi* infection mainly occurs through contact with contaminated feces of hematophagous triatomine bugs, which are the vectors of this parasite [4]. In addition, the transmission may occur congenitally or by blood transfusions and organ transplants, representing the main modes of infection in urban areas and non-endemic countries. Moreover, *T. cruzi* may be transmitted through the consumption of food contaminated with the parasite [5].

Once the parasite is acquired, Chagas disease involves acute and chronic clinical phases. The acute phase usually goes unnoticed due to non-specific symptoms and generally lasts 4 to 8 weeks [6]. However, during the chronic phase, which may develop 10 to 30 years after infection, an estimated 30% to 40% of infected individuals develop potentially lethal cardiac or gastrointestinal disease [6,7].

Chagas disease becomes chronic without treatment, and only the drugs nifurtimox and benznidazole, developed more than 50 years ago, are commercially available for its treatment [3]. These drugs show more than 80% efficacy during the acute phase [5]. Nevertheless, they have limitations. They are marginally effective during the chronic phase, require long treatment periods, and produce side effects [8]. Benznidazole commonly generates dermatitis, peripheral neuropathy, anorexia, bone marrow suppression, nausea, and vomiting, whereas nifurtimox causes anorexia, nausea, vomiting, abdominal pain, headache, dizziness, and neuropathy. As a result, many patients discontinue treatment [1]. Furthermore, these drugs are cytotoxic and genotoxic [9]. In addition, some *T. cruzi* strains have been reported to be naturally resistant to both drugs, making them ineffective [10]. Therefore, it is of utmost importance to seek alternative treatments.

The development of new antiparasitic drugs has not been a priority for the pharmaceutical industry because many parasitic diseases occur in poor countries, where people cannot afford high-priced medications [11]. An alternative approach involves exploring plant extracts or their secondary metabolites for their antiparasitic properties. Plants are a source of a wide range of natural products that possess various therapeutic properties and are continuously explored to develop novel drugs [12]. Nowadays, more than 25% of drugs used during the last 20 years are directly derived from plants or are chemically altered molecules. However, only 5% to 15% of the approximately 260,000 higher plants have been investigated for bioactive compounds [13].

*Schinus molle* L. (1753) commonly known as pepper tree or pirul, is an evergreen tree from the Anacardiaceae family that originates in South America and is widespread around the globe, including the Mediterranean, tropical, and subtropical regions, as well as South Africa [14]. Traditionally, this plant has been used to treat cough, tuberculosis, bronchitis, fever, eye infection, allergy, hemorrhoids, respiratory infections, jaundice, diarrhea, and tonsillitis. In Ethiopia, this plant is also used to treat malaria [15]. Furthermore, its crude methanol extract has been shown to possess anti-*T. cruzi* activity [16,17]. However, its active compounds have not been identified yet. Therefore, we aimed to evaluate the anti-*T. cruzi* potential of different *S. molle* extracts, partitions, and fractions and identify their active compounds.

## 2. Results

Figure 1 shows the general experimental protocol implemented during the bioguided fractionation of *S. molle* crude methanol extract, as well as the IC_50_ against *T. cruzi* obtained for each extract.

### 2.1. Biological Activity of S. molle Crude Methanol Extract and Partitions

We evaluated *S. molle* crude methanol extract (Sm-ME) and partitions obtained with solvents of increasing polarity (*n*-hexane, chloroform, and methanol). The yield and biological activity of these extracts are shown in Table 1. Sm-ME showed a yield of 21.5%, with an anti-*T. cruzi* IC_50_ of 94 µg/mL and an SI of 2.3. The partition with the highest yield was Sm-MP (42.4%, 2.4 g), followed by Sm-HP (24.8%, 1.4 g) and Sm-CP (8.3%, 0.5 g). It is important to mention that after continuous Soxhlet extractions, an insoluble fraction was obtained with a significant yield (19.9%, 1.1 g), which was not evaluated because of its insolubility. Furthermore, despite presenting the lowest yield, Sm-CP was the one that showed the highest anti-*T. cruzi* activity with an IC_50_ of 20 µg/mL and an SI of 5.6, whereas the partition with the highest yield (Sm-MP) did not show biological activity. We observed that Sm-CP showed better activity than nifurtimox, which exhibited an IC_50_ of 32 µg/mL. Based on their IC_50_ values, Sm-ME and Sm-CP were moderately cytotoxic, whereas Sm-HP was cytotoxic on Vero cells; Sm-HP and Sm-CP were moderately toxic, whereas Sm-ME was not toxic to *A. salina*, and all extracts were non-hemolytic (Table 1).

### 2.2. Activity of S. molle Hexane and Chloroform Fractions

Five collective fractions of Sm-HP (Sm-HF1 to Sm-HF5) and Sm-CP (Sm-CF1 to Sm-CF5) were obtained, showing yields from 2.7% to 66.9%. The yield and biological activity of the fractions are shown in Table 2. SmHF1 presented the highest anti-*T. cruzi* activity with an IC_50_ of 16 µg/mL and an SI of 5.8 and Sm-CF3 showed the highest activity with an IC_50_ of 21 µg/mL and an SI of 6.8, having the highest SI of the ten fractions. Based on their IC_50_ values, all fractions evaluated were classified as moderately cytotoxic except for Sm-HF1, which was cytotoxic on Vero cells. All fractions were moderately toxic except for Sm-HF2 and Sm-CF4, which were not toxic on *A. salina*, and all were non-hemolytic (Table 2).

### 2.3. Identified Compounds in the S. molle Sm-CF3 Fraction

Analysis of the Sm-CF3 fraction with GC-MS revealed the presence of 20 compounds (Table 3, Figure 2, and Appendix A). Most of them belong to the chemical class of fatty acyls (11/20 = 55%), followed by organometalloid compounds (2/20 = 10%), prenol lipids (2/20 = 10%), and benzene and substituted derivatives, organooxygen compounds, steroids and steroid derivatives, acyl halides, and hydroxy acids and derivatives (1/20 = 10% each one). These compounds have different pharmacological effects but most of these reports evaluated antimicrobial activity, followed by antiparasitic, and antifungal activity. Compounds with antiparasitic activity reported were cis-5,8,11,14,17-eicosapentaenoic acid and trans-Z-α-bisabolene epoxide (Figure 3).

## 3. Discussion

In this study, the pharmacological potential of *S. molle* against *T. cruzi* was evaluated. Although there are reports of its anti-*T. cruzi* activity [16,17], only the crude methanol extract has been investigated. However, identification of anti-*T. cruzi* compounds and their toxicity and SI remain to be elucidated. According to Osorio et al. classification [18], most evaluated extracts fall into the category of active (IC_50_ between 10 µg/mL to 50 µg/mL), with Sm-HF1 being the most potent with an IC_50_ of 16 µg/mL, as compared with nifurtimox activity (IC_50_ = 32 µg/mL). Ten-fold differences among the IC_50_ values of Sm-ME and SM-MP were obtained. This may be due to the crude methanol extract (Sm-ME), which contains all non-polar, moderately polar, and polar compounds that were extracted from the plant. Furthermore, in the methanol partition (Sm-MP), only polar compounds of the plant are present, since non-polar and moderately polar compounds were extracted in the hexane and chloroform partitions, respectively. This indicates that the compounds with activity against *T. cruzi* are mainly non-polar and moderately polar compounds, as confirmed by their identification. This confirms the previously reported anti-*T. cruzi* activity of the plant. Furthermore, this plant has shown activity against other protozoa such as *Leishmania amazonensis* [19] and *Plasmodium berghei* [14], as well as helminthicide activity against *Haemonchus contortus* [20]. This underscores its broad-spectrum antiparasitic potential, which makes this plant an ideal candidate to identify compounds with antiparasitic activity.

Moreover, the *T. cruzi* strain used in this study belongs to the discrete typing unit TcI, which is the most prevalent in Latin America and known for its resistance to conventional antichagasic drugs [21,22]. Our findings indicate significant activity of *S. molle* against this strain, suggesting its potential as an alternative or adjunct therapy for Chagas disease, particularly in regions with prevalent drug-resistant strains.

In addition, Vero cells were selected to evaluate cytotoxicity and determine the SIs of the extracts, as they are commonly used for such evaluations in studies investigating extracts of plants for antiparasitic activity [23]. According to Osorio et al.’s classification [18], most of the extracts were classified as moderately cytotoxic (IC_50_ between 100 µg/mL to 1000 µg/mL), as compared with the well-known cytotoxicity of nifurtimox and benznidazole [9]. For parasites, a plant extract may be assumed bioactive and non-toxic if SI > 1, indicating differing toxic and parasitic components [24]. The higher the SI value, the safer the extract. Most evaluated extracts in this study have SIs > 2, indicating optimal antiparasitic activity, with Sm-CF3 showing the highest SI of 6.8, suggesting promising activity.

Furthermore, the *Artemia salina* assay serves as a preliminary toxicity assessment tool [25] and correlates strongly with acute oral toxicity results in mice, and according to Fernández-Calienes et al. classification [26], most of the extracts are moderately toxic (IC_50_ between 100 µg/mL to 1000 µg/mL), indicating promising results, as they demonstrated adequate anti-*T. cruzi* activity and low toxicity [26].

Since Sm-CF3 showed the highest SI, we identified its constituent compounds. Most of the identified compounds belong to the chemical class of fatty acyls (55%). It has been previously reported that fatty acids possess important biological properties such as antibacterial, antifungal, and antiparasitic activity [27]. Therefore, this group may be responsible for the anti-*T. cruzi* activity of the fraction.

Among the 20 compounds identified, two of them have been reported as antiparasitic. However, they have not been assessed in their pure form. Cholestan-3-ol, 2-methylene- (3β,5α) has been identified in *Achillea wilhelmsii* extract, which shows activity against *Leishmania major* [28], whereas isopinocarveol has been identified in *Melaleuca styphelioides* extract, showing activity against *Acanthamoeba castellanii* [29]. In addition, other compounds have been evaluated in their pure form, demonstrating antiparasitic activity. Cis-5,8,11,14,17-eicosapentaenoic acid has activity against *Trichomonas vaginalis* [30], whereas trans-Z-α-bisabolene epoxide has activity against *Leishmania* sp. [31]. As *Leishmania* and *Trypanosoma* are very similar protozoa, with them being hemoflagellates and belonging to the Trypanosomatidae family [32], trans-Z-α-bisabolene epoxide may have anti-*T. cruzi* potential.

Nevertheless, further biotargeted fractionation is required to isolate and identify the bioactive compounds with antiparasitic activity. Moreover, it is essential to evaluate these compounds in an in vivo murine model to validate their antiparasitic efficacy and to exclude any potential toxicity to humans.

## 4. Materials and Methods

### 4.1. Ethical Statement

The procedures used in this study were approved by the Ethics Committee of the Facultad de Ciencias Biológicas (FCB) at the Universidad Autónoma de Nuevo León (UANL), registration no. CI-08-2020. Experiments involving human erythrocytes were conducted with the informed consent of a healthy donor in compliance with the Official Mexican Technical Standard NOM-253-SSA1-2012 [33].

### 4.2. Plant Material

*Schinus molle* L. was collected in June 2020 in Cadereyta Jiménez, Nuevo León, México (N 25°32′17.079″; W 99°56′52.457″). Taxonomic identification of the plant was performed at the herbarium of the FCB-UANL, with a voucher number 030594. The taxonomic validation of the name and family of the plant was performed using the International Plant Names Index (https://www.ipni.org/; accessed on 20 April 2024).

### 4.3. Preparation of Extracts

Leaves and stems of *S. molle* were dried at room temperature, after which they were powdered with a manual grinder. The crude methanol extract (Sm-ME) was obtained via Soxhlet extraction. For this, 25 g of the plant was placed in a Soxhlet extractor with 500 mL of absolute methanol (CTR Scientific, Monterrey, NL, Mexico). Extraction was sustained for 48 h, after which the extract was filtered and concentrated under reduced pressure at 40 °C with a rotary evaporator (Yamato Digital Rotary Evaporator, RE301, Santa Clara, CA, USA). The residual solvent was evaporated at room temperature with a vacuum desiccator [34]. Next, hexane (Sm-HP), chloroform Sm-CP), and methanol (Sm-MP) partitions were obtained via continuous Soxhlet extractions of the Sm-ME. For this, we used five grams of the Sm-ME and 250 mL of *n*-hexane, chloroform, and methanol as extraction solvents. Each extraction was maintained for 48 h, after which fractions were filtered and concentrated as performed for the Sm-ME [35].

Extract yield was calculated as previously reported [36]. Next, 25 mg of each extract was solubilized in one milliliter of dimethyl sulfoxide (DMSO; Sigma-Aldrich, St. Louis, MO, USA) and stored at 4 °C until use. The final concentration of DMSO used in the experiments was less than 1% (*v*/*v*), which did not alter cell and parasite viability.

### 4.4. Anti-Trypanosoma Cruzi Activity

The *T. cruzi* NL strain was originally isolated from *Triatoma gerstaeckeri* collected in Nuevo León, México [17]. *T. cruzi* epimastigotes were cultured in Liver Infusion Tryptose (LIT) medium supplemented with 10% fetal bovine serum (FBS; Gibco, Grand Island, NY, USA) at 28 °C and harvested during the exponential growth phase, with a cell density of approximately 1 × 10^6^ parasites/mL [16].

We placed 1 × 10^6^ parasites/well in round-bottomed 96-well microplates (Corning Incorporated, Corning, NY, USA) in LIT medium. Parasites were treated with 10 μg/mL to 1000 μg/mL of extracts. We used 35 µg/mL nifurtimox (Sigma-Aldrich) as a positive control and untreated culture medium with parasites as a negative control. We also used as a control 1% DMSO. Microplates were incubated for 72 h at 28 °C, and parasite viability was determined using the 3-[4,5-dimethylthiazol-2-yl]-2,5-diphenyl tetrazolium bromide (MTT; Affymetrix, Cleveland, OH, USA) colorimetric assay, as previously reported [37]. *T. cruzi* percentage growth inhibition was calculated as previously reported [36]. Extracts were classified as highly active (IC_50_ < 10 µg/mL), active (IC_50_ > 10 < 50 µg/mL), moderately active (IC_50_ > 50 < 100 µg/mL), and non-active (IC_50_ > 100 µg/mL) [18].

### 4.5. Cytotoxic Activity in Vero Cells

African green monkey kidney epithelial cells (Vero; ATCC CCL-81) were grown in RPMI-1640 culture medium (Gibco) supplemented with 10% heat-inactivated FBS (Gibco), 2 g/L sodium bicarbonate (NaHCO_3_), and 1% penicillin/streptomycin solution (Gibco) (referred to as complete RPMI medium). Cells were cultured at 37 °C in an atmosphere of 5% CO_2_ in air [38].

Vero cells were seeded at a concentration of 2.2 × 10^4^ cells/well in flat-bottomed 96-well microplates (Corning Incorporated) in complete RPMI medium and incubated 24 h before treatment. Next, the cells were exposed to 10 μg/mL to 1000 μg/mL of extracts for 24 h. The controls were culture medium alone and 1% DMSO. Cell viability was determined with the MTT colorimetric assay by adding 20 μL of MTT (5 mg/mL) to each well and incubating them for one hour. The plates were then decanted and formazan crystals were dissolved with 100 µL of DMSO. Optical densities (ODs) were determined at 570 nm using a microplate reader (ELISA Variouskan LUX multimode; Thermo Fisher Scientific, Waltham, MA, USA) [39]. Percentage growth inhibition was calculated as previously reported [20] and extracts were classified as highly cytotoxic (IC_50_ < 10 µg/mL), cytotoxic (IC_50_ > 10 < 100 µg/mL), moderately cytotoxic (IC_50_ > 100 < 1000 µg/mL), and potentially non-cytotoxic (IC_50_ > 1000 µg/mL) [18].

### 4.6. Determination of Selectivity Indices of Extracts

In this study, Vero cells were used as a mammalian cell model for testing the unspecific cytotoxicity of the extracts. Extract selectivity indices (SIs) were calculated by dividing the IC_50_ of Vero cells by the IC_50_ of *T. cruzi* using the following formula [38]:$$\mathrm{SI}=\frac{{\mathrm{IC}}_{50}\;\mathrm{Vero}\;\mathrm{cells}\;}{{\mathrm{IC}}_{50}\;T.\;cruzi}$$

### 4.7. Toxic Activity in Artemia Salina

*Artemia salina* eggs (INVE Aquaculture NV, Basrode, Belgium) were hatched in a microaquarium with 4 L of 3.7% saline solution at pH 8, suitably oxygenated, and at 25 °C to 30 °C under continuous light. Water was supplemented with 0.75 g/L of yeast extract. Eggs hatched in 24 h and we obtained larvae (nauplii). At 48 h after hatching, the larvae were transferred to flat-bottomed 24-well microplates at 10 larvae/well, and 10 μg/mL to 1000 μg/mL of the extracts was evaluated for 24 h at 25 °C to 30 °C under continuous light. We used 100 μg/mL potassium dichromate (K_2_Cr_2_O_7_) as a positive control and saline solution as a negative control; we also used as a control 1% DMSO. Larvae survival was evaluated as previously reported [25]. Results were considered valid if the percentage of mortality in the negative controls did not exceed 10% [40].

Extracts were classified as highly toxic (IC_50_ < 10 µg/mL), toxic (IC_50_ > 10 < 100 µg/mL), moderately toxic (IC_50_ > 100 < 1000 µg/mL), and non-toxic (IC_50_ > 1000 µg/mL) [26].

### 4.8. Human Erythrocyte Hemolytic Activity Assay

Blood from healthy donors (20 mL) was deposited in tubes containing the anticoagulant EDTA (BD Diagnostics, Franklin Lakes, NJ, USA). Red blood cells were washed three times with phosphate-buffered saline solution (PBS; pH 7.2) and a 5% erythrocytes suspension was prepared in sterile PBS. Next, 200 µg/mL to 1000 µg/mL extracts and 5% erythrocytes suspension were incubated for 30 min at 37 °C. Distilled water was used as a positive control for hemolysis and PBS as a negative control; we also used as a control 1% DMSO. After incubation, the samples were centrifuged at 13,000 for five minutes at 4 °C, after which 200 µL of the supernatant from each tube was transferred to a flat-bottomed 96-well microplate to measure the OD at 540 nm of the released hemoglobin in a microplate reader (ASYS UVM; Biochrom Ltd., Cambridge, UK). The percentage of hemolysis for each sample was calculated as previously reported [34].

### 4.9. Column Chromatography Fractionation

As Sm-HP and Sm-CP partitions showed the highest anti-*T. cruzi* activity, they were fractionated using column chromatography. For this, open glass chromatographic columns (330 mm × 28 mm) were prepared with 20 g of 0.063 to 0.200 mm particle size silica gel 60 G (Merck, Darmstadt, Germany) [41]. For Sm-HP, 500 mg of the partition was added to the column and eluted with stepwise gradients of 20 mL of *n*-hexane-chloroform, chloroform-ethyl acetate, and ethyl acetate-methanol at 100:0, 90:10, 80:20, 70:30, 60:40, 50:50, 40:60, 30:70, 20:80, 10:90, and 0:100 *v*/*v* stepwise gradient proportions. We then obtained 124 fractions of five milliliters and pooled them based on their thin layer chromatography (TLC) profile (hexane-chloroform 1:1) to yield five collective fractions (Sm-HF1 to Sm-HF5). For Sm-CP, 500 mg of the partition was added to the column and eluted with stepwise gradients of chloroform–ethyl acetate and ethyl acetate–methanol as performed by Sm-HP. We then obtained 84 fractions of five milliliters and pooled them based on their TLC profile (chloroform-ethyl acetate 1:1) to yield five collective fractions (Sm-CF1 to Sm-CF5). The anti-*T. cruzi*, cytotoxic, toxic, and hemolytic activity of the fractions were evaluated using the methods mentioned above.

### 4.10. Identification of Bioactive Compound Using Gas Chromatography–Mass Spectrometry (GC-MS)

We selected the fraction with the highest anti-*T. cruzi* activity (Sm-CF3) to identify its constituent compounds using GC-MS in the external services laboratory of the Instituto Politécnico Nacional (IPN) in México City.

For the analysis, we used a gas chromatograph (Agilent Technologies 7890A; Santa Clara, CA, USA) equipped with an Agilent Technologies 5975C triple mass detector and an Agilent 123-2332DB-23 column (60 m × 320 µm × 0.25 µm). The operating conditions were as follows: the initial oven temperature was set at 170 °C for two minutes and then increased by 5 °C/min until reaching 250 °C, where it was kept for 15 min. The injector temperature was maintained at 250 °C. Helium (99.999%) was used as the carrier gas with a flow rate of one milliliter per minute, and the injection volume was two microliters. The GC–MS mass spectrum data were analyzed and identified using the National Institute Standard and Technology (NIST) database to determine their *m*/*z* values according to their 100% abundance [42]. The chemical classes of the identified metabolites were automatically determined using the Classyfire web-based application (http://classyfire.wishartlab.com, accessed on 10 March 2023) [43].

### 4.11. Statistical Analysis

Data represent the mean ± SD of at least triplicate determinations and three independent experiments, with a confidence level of 95%. The Probit test was used to calculate the IC_50_ (half maximal inhibitory concentration) values. A one-way analysis of variance was used to determine the significant difference between the tested extracts and Tukey’s post hoc test was used to determine the difference between the treatment means. Statistical analyses were performed using IBM SPSS Statistics v25.0 software (IBM Corp., Armonk, NY, USA).

## 5. Conclusions

This study demonstrates that the crude methanol extract, partitions, and fractions of *S. molle* exhibit anti-*T. cruzi* activity against the NL strain. In particular, Sm-CF3 showed promising activity (IC_50_ = 19 µg/mL; SI 6.8). In this fraction, we identified 20 compounds via GC-MS, among which the antiparasitic activity of cis-5,8,11,14,17-eicosapentaenoic acid and trans-Z-α-bisabolene epoxide has been reported, suggesting their potential role in the observed anti-*T. cruzi* activity.

Our findings support the therapeutic potential of *S. molle* as a source of novel antiparasitic agents against *T. cruzi*. However, further research is warranted to validate the efficacy of identified compounds in preclinical models and ultimately advance them toward clinical trials for the treatment of Chagas disease.

## Figures and Tables

**Figure 1 plants-13-02177-f001:**
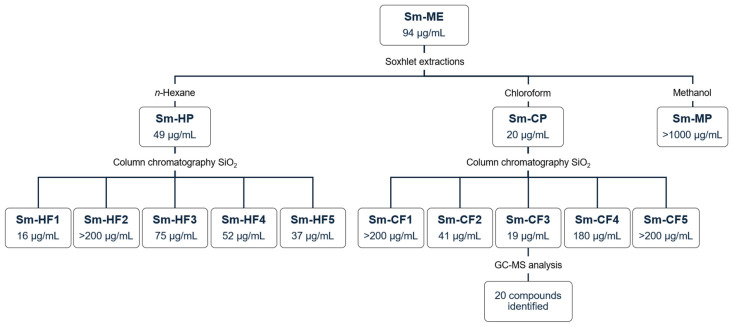
Bioguided fractionation of *S. molle*. Data represent IC_50_ against *T. cruzi*. Sm-ME: *S. molle* crude methanol extract; Sm-HP: *S. molle n*-hexane partition; Sm-CP: *S. molle* chloroform partition; Sm-MP: *S. molle* methanol partition; Sm-HF: *S. molle* hexane fraction; Sm-CF: *S. molle* chlorofom fraction.

**Figure 2 plants-13-02177-f002:**
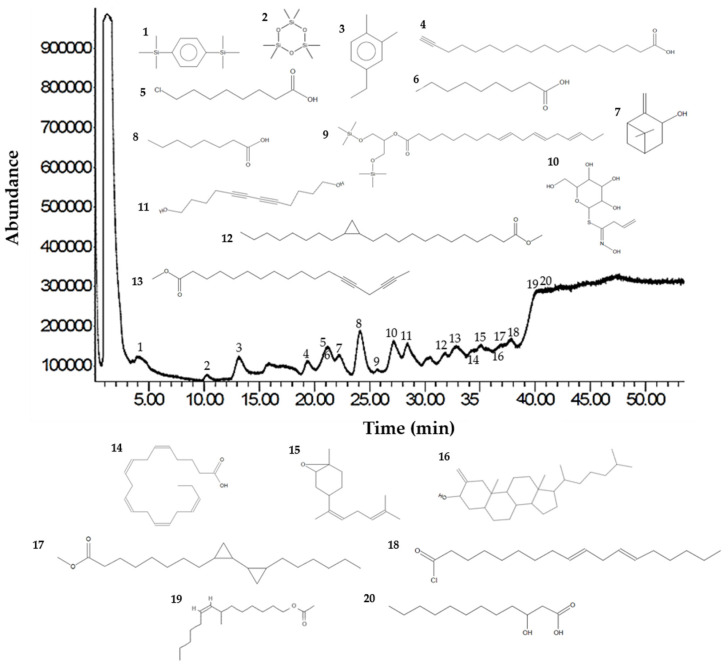
GC-MS chromatogram of the Sm-CF3 fraction. The UPAC names, synonym, formula, mass, *m*/*z*, and class of the compounds are explained in Table 3.

**Figure 3 plants-13-02177-f003:**
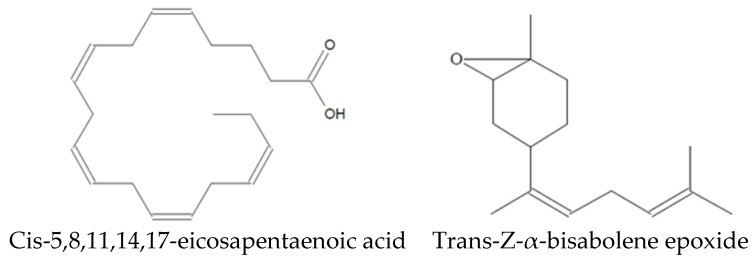
Compounds identified in Sm-CF3 with reported antiparasitic activity.

**Table 1 plants-13-02177-t001:** Yield, anti-*Trypanosoma cruzi*, cytotoxic, toxic, and hemolytic activity of *S. molle* crude methanol extract and partitions.

Extract	% Yield	IC_50_ (µg/mL)				SI
		*T. cruzi*	Vero	*A. salina*	Erythrocytes	
Sm-ME	21.5	94 ± 7.1 ^c^	212 ± 8.1 ^c^	>1000	163 ± 4.7 ^a 2^	2.3
Sm-HP	24.8	49 ± 5.3 ^b^	85 ± 5.6 ^a^	632 ± 3.2 ^b^	235 ± 5.2 ^c 1^	1.7
Sm-CP	8.3	20 ± 3.2 ^a^	112 ± 7.8 ^b^	581 ± 4.1 ^a^	202 ± 5.1 ^b 1^	5.6
Sm-MP	42.4	>1000 ^†^	>1000 ^†^	>1000 ^†^	>1000 ^†^	ND

IC_50_ values (µg/mL) are the means ± SD, with significant (*p* < 0.05) differences indicated by different letters within the same column, as determined using the post hoc Tukey’s test. SI represents the IC_50_ of Vero cells divided by the IC_50_ of *T. cruzi*. ^†^ As the IC_50_ was above 1000 µg/mL, these values were not considered for Tukey’s analysis. ND: Not determined because this extract did not show activity at any of the evaluated concentrations. ^1^ This extract, at the IC_50_ required to kill *T. cruzi*, did not cause significant hemolysis. ^2^ This extract, at the IC_50_ required to kill *T. cruzi*, caused low hemolysis (<20%).

**Table 2 plants-13-02177-t002:** Yield, anti-*Trypanosoma cruzi*, cytotoxic, toxic, and hemolytic activity of *S. molle* hexane and chloroform fractions.

Fraction	% Yield	IC_50_ (µg/mL)				SI
		*T. cruzi*	Vero	*A. salina*	Erythrocytes	
Sm-HF1	2.1	16 ± 2.6 ^a^	93 ± 7.2 ^a^	528 ± 6.7 ^b^	198 ± 6.2 ^b 1^	5.8
Sm-HF2	2.7	>200 ^†^	144 ± 8.3 ^c^	>1000 ^†^	>1000 ^†^	ND
Sm-HF3	10.1	75 ± 5.6 ^d^	336 ± 9.1 ^f^	743 ± 8.2 ^e^	>1000 ^†^	4.5
Sm-HF4	66.9	52 ± 6.3 ^c^	173 ± 5.3 ^e^	632 ± 5.1 ^c^	354 ± 4.3 ^e 1^	3.3
Sm-HF5	10.8	37 ± 5.1 ^b^	147 ± 4.1 ^c^	681 ± 7.9 ^d^	237 ± 4.7 ^c 1^	4.0
Sm-CF1	3.7	>200 ^†^	NE	NE	NE	NE
Sm-CF2	3.1	41 ± 6.3 ^b^	NE	NE	NE	NE
Sm-CF3	32.1	19 ± 3.7 ^a^	130 ± 5.4 ^b^	428 ± 9.8 ^a^	161 ± 5.2 ^a 1^	6.8
Sm-CF4	27.4	180 ± 4.9 ^e^	156 ± 6.2 ^d^	>1000 ^†^	>1000 ^† 1^	0.9
Sm-CF5	29.0	>200 ^†^	330 ± 5.9 ^f^	513 ± 10.1 ^b^	311 ± 6.8 ^d^	ND

IC_50_ values (µg/ML) are shown as the means ± SD, with significant (*p* < 0.05) differences indicated by different letters within the same column, as determined using the post hoc Tukey’s test. SI represents the IC_50_ of Vero cells divided by the IC_50_ of *T. cruzi*. ^†^ These values were not considered for Tukey’s analysis. ND: Not determined because this fraction did not show activity against *T. cruzi* at any of the evaluated concentrations. NE: Not evaluated because this fraction has a low yield. ^1^ This extract, at the IC_50_ required for *T. cruzi*, did not cause significant hemolysis.

**Table 3 plants-13-02177-t003:** GC-MS results of the Sm-CF3 fraction.

No.	RT	UPAC Name	Synonym	Formula	Mass	*m*/*z*	Class
1	4.0	Trimethyl-(4-trimethylsilylphenyl) silane	1,4-Bis (trimethylsilyl) benzene	C_12_H_22_Si_2_	222.47	207	OMC
2	10.2	2,2,4,4,6,6-hexamethyl-1,3,5,2,4,6-trioxatrisilinane	Hexamethylcyclotrisiloxane	C_6_H_18_O_3_Si_3_	222.46	207	OMC
3	13.1	4-ethyl-1,2-dimethylbenzene	Benzene, 4-ethyl-1,2-dimethyl-	C_10_H_14_	134.22	119	BSD
4	19.0	Octadec-17-ynoic acid	17-Octadecynoic acid	C_11_H_20_O_2_	280.4	55	FA
5	21.2	8-chlorooctanoic acid	8-Chlorocapric acid	C_8_H_15_ClO_2_	178.65	60	FA
6	21.2	Nonanoic acid	Pelargonic acid	C_9_H_18_O_2_	158.24	60	FA
7	22.3	(1*R*,3*R*,5*R*)-6,6-dimethyl-2-methylidenebicyclo[3.1.1]heptan-3-ol	Isopinocarveol	C_10_H_16_O	152.23	41	PL
8	24.1	Octanoic acid	Caprylic acid	C_8_H_16_O_2_	144.21	60	FA
9	25.7	1,3-bis (trimethylsilyloxy) propan-2-yl (9*E*,12*E*,15*E*)-octadeca-9,12,15-trienoate	9,12,15-Octadecatrienoic acid, 2- [(trimethylsilyl) oxy]-1-[(trimethylsilyl) oxy]methyl] ethyl ester, (Z,Z,Z)-	C_27_H_52_O_4_Si_2_	496.9	41	FA
10	27.2	[3,4,5-trihydroxy-6-(hydroxymethyl) oxan-2-yl] (1*E*)-*N*-hydroxybut-3-enimidothioate	Desulphosinigrin	C_10_H_17_NO_6_S	279.31	60	OOX
11	28.4	Dodeca-5,7-diyne-1,12-diol	5,7-Dodecadiyn-1,12-diol	C_12_H_18_O_2_	194.27	91	FA
12	31.9	Methyl 12-(2-octylcyclopropyl) dodecanoate	Cyclopropanedodecanoic acid, 2-octyl-,methyl ester	C_24_H_46_O_2_	366.6	41	FA
13	32.9	Methyl octadeca-13,16-diynoate	13,16-Octadecadiynoic acid, methyl ester	C_19_H_30_O_2_	290.4	74	FA
14	34.3	(5*E*,8*E*,11*E*,14*E*,17*E*)-icosa-5,8,11,14,17-pentaenoic acid	Cis-5,8,11,14,17-eicosapentaenoic acid	C_20_H_30_O_2_	302.5	79	FA
15	35.1	(3*R*,5*S*,8*R*,9*S*,10*S*,13*R*,14*S*,17*R*)-10,13-dimethyl-17-[(2*R*)-6-methylheptan-2-yl]-2-methylidene-1,3,4,5,6,7,8,9,11,12,14,15,16,17-tetradecahydrocyclopenta[a]phenanthren-3-ol	Cholestan-3-ol, 2-methylene-, (3β,5α)	C_28_H_48_O	400.7	69	SSD
16	37.0	1-methyl-4-[(2*Z*)-6-methylhepta-2,5-dien-2-yl]-7-oxabicyclo[4.1.0]heptane	Trans-Z-α-bisabolene epoxide	C_15_H_24_O	220.35	43	PL
17	36.9	Methyl 8-[2-(2-hexylcyclopropyl) cyclopropyl]octanoate	[1,1′-Bicyclopropyl]-2-octanoic acid, 2′-hexyl-, methyl ester	C_21_H_38_O_2_	322.5	73	FA
18	37.8	Octadeca-9,12-dienoyl chloride	9,12-Octadecadienoyl chloride, (Z,Z)-	C_18_H_31_ClO	298.9	55	AH
19	40.1	3-hydroxydodecanoic acid	Dodecanoic acid, 3-hydroxy	C_12_H_24_O_3_	216.32	43	HAD
20	41.2	[(*Z*)-7-methyltetradec-8-enyl] acetate	7-Methyl-Z-tetradecen-1-ol acetate	C_17_H_32_O_2_	268.4	55	FA

RT: retention time (min); OMC: organometalloid compounds; BSD: benzene and substituted derivatives; FA: fatty acyls; PL: prenol lipids; OOC: organooxygen compounds; SSD: steroids and steroid derivatives; AH: acyl halides; HAD: hydroxy acids and derivatives.

## Data Availability

Data are contained within the article.

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
