# Peer review of "In Vitro Antiprotozoal Activity of Schinus molle Extract, Partitions, and Fractions against Trypanosoma cruzi"

_plants, 2024, doi:10.3390/plants13162177_

Round 1
Reviewer 1 Report
Comments and Suggestions for Authors
The topic presented is of great relevance in the area of ​​study of Chagas Disease. The drug used is Benznidazole, which does not have an excellent percentage of accuracy, and at the same time presents different side effects such as dermatitis, peripheral neuropathy, anorexia, bone marrow suppression, nausea, and vomiting, whereas nifurtimox causes anorexia, nausea, vomiting , abdominal pain, headache, dizziness, and neuropathy. As a result, many patients discontinue the treatment.
a simple problem that was characterized is the fact that the authors called the supplementary figure figure 1. This is problematic, it was difficult even for the reviewer to understand that there is no figure 1 in the text. So please change the name of the figure in the text to supplementary figure.
Another issue is the lack of graphs in the MS. The authors could present the result as a graph, it would be easier for the reader to understand the result presented.
English is ok and the manuscript is interesting.
Author Response
Review Report Form
Reviewer 1
Open Review
(x) I would not like to sign my review report
( ) I would like to sign my review report
Quality of English Language
(x) I am not qualified to assess the quality of English in this paper
( ) English very difficult to understand/incomprehensible
( ) Extensive editing of English language required
( ) Moderate editing of English language required
( ) Minor editing of English language required
( ) English language fine. No issues detected
|
Yes |
Can be improved |
Must be improved |
Not applicable |
|
|
Does the introduction provide sufficient background and include all relevant references? |
(x) |
( ) |
( ) |
( ) |
|
Is the research design appropriate? |
(x) |
( ) |
( ) |
( ) |
|
Are the methods adequately described? |
(x) |
( ) |
( ) |
( ) |
|
Are the results clearly presented? |
( ) |
( ) |
(x) |
( ) |
|
Are the conclusions supported by the results? |
(x) |
( ) |
( ) |
( ) |
Comments and Suggestions for Authors
The topic presented is of great relevance in the area of ​​study of Chagas Disease. The drug used is Benznidazole, which does not have an excellent percentage of accuracy, and at the same time presents different side effects such as dermatitis, peripheral neuropathy, anorexia, bone marrow suppression, nausea, and vomiting, whereas nifurtimox causes anorexia, nausea, vomiting , abdominal pain, headache, dizziness, and neuropathy. As a result, many patients discontinue the treatment.
A simple problem that was characterized is the fact that the authors called the supplementary figure figure 1. This is problematic, it was difficult even for the reviewer to understand that there is no figure 1 in the text. So please change the name of the figure in the text to supplementary figure.
Response: Thank you very much for your comment. It was an oversight not to include Figure 1 in the document. It has now been added (Lines 84 to 86 of the revised manuscript). In addition, complete names of abbreviations were cited (Lines 86 to 88 of the revised manuscript).
Figure 1. Bioguided fractionation of S. molle. Data represent IC50 against T. cruzi. Sm-ME: S. molle crude methanol extract; Sm-HP: S. molle n-hexane partition; Sm-CP: S. molle chloroform partition; Sm-MP: S. molle methanol partition; Sm-HF: S. molle hexane fraction; Sm-CF: S. molle chlorofom fraction.
Another issue is the lack of graphs in the MS. The authors could present the result as a graph, it would be easier for the reader to understand the result presented.
Response: This issue was considered, and a graph was included in the revised manuscript (lines 139 and 140) as shown below:
Figure 2. GC-MS chromatogram of Sm-CF3 fraction. UPAC names, Synonym, Formula, Mass, m/z, and Class of the compounds are explained in Table 3.
English is ok and the manuscript is interesting.

Reviewer 2 Report
Comments and Suggestions for Authors
The study by Garza et. al. is well-designed and shows the potential of S. mole extract in the treatment of Chagas disease. I have the following comments/questions before accepting the manuscript for publication:
Please add a reference to recent important reviews such as
de Sousa, A.S., Vermeij, D., Ramos, A.N. and Luquetti, A.O., 2024. Chagas disease. The Lancet, 403(10422), pp.203-218.
Martín-Escolano, J., Marín, C., Rosales, M.J., Tsaousis, A.D., Medina-Carmona, E. and Martín-Escolano, R., 2022. An updated view of the Trypanosoma cruzi life cycle: Intervention points for an effective treatment. ACS Infectious Diseases, 8(6), pp.1107-1115.
The key for short forms (e.g. Sm-ME, Sm-CP) used in Table 1 must be clearly specified at the end of Table. Also, mention the long forms for Sm-HF1, etc.
Can authors comment/explain > 10-fold difference in the IC50 values of Sm-ME and SM-MP? They should have the same content in principle, so this result is confusing.
Can authors report Nifurtimox data in the same table? What was SI for Nifurtimox in this study?
It is important to include some examples of raw data used to calculate IC50 values in this study. Include one example of control (nifurtimox) and 1-2 examples of the extracts.
How were the compounds identified after GC-MS? Did authors use any specific database to predict based on m/z? Can authors comment on the relative abundance of the 20 compounds identified?
The structure of eicosapentaenoic acid needs to be corrected in Figure 2.
Author Response
REVIEWER 2
Open Review
(x) I would not like to sign my review report ( ) I would like to sign my review report
Quality of English Language
( ) I am not qualified to assess the quality of English in this paper ( ) English very difficult to understand/incomprehensible ( ) Extensive editing of English language required ( ) Moderate editing of English language required ( ) Minor editing of English language required (x) English language fine. No issues detected
Yes
Can be improved
Must be improved
Not applicable
Does the introduction provide sufficient background and include all relevant references?
( )
(x)
( )
( )
Is the research design appropriate?
(x)
( )
( )
( )
Are the methods adequately described?
( )
(x)
( )
( )
Are the results clearly presented?
( )
(x)
( )
( )
Are the conclusions supported by the results?
(x)
( )
( )
( )
Comments and Suggestions for Authors
The study by Garza et. al. is well-designed and shows the potential of S. mole extract in the treatment of Chagas disease. I have the following comments/questions before accepting the manuscript for publication:
Please add a reference to recent important reviews such as
de Sousa, A.S., Vermeij, D., Ramos, A.N. and Luquetti, A.O., 2024. Chagas disease. The Lancet, 403(10422), pp.203-218.
Martín-Escolano, J., Marín, C., Rosales, M.J., Tsaousis, A.D., Medina-Carmona, E. and Martín-Escolano, R., 2022. An updated view of the Trypanosoma cruzi life cycle: Intervention points for an effective treatment. ACS Infectious Diseases, 8(6), pp.1107-1115.
Response: Regarding your observation, references were included in lines 33 to 35; 47 and 48, and 51 and 52, as follows:” … is a parasitic zoonotic disease caused by the hemoflagellate
protozoan Trypanosoma cruzi (Chagas, 1909) [1]. It is the most important parasitic disease in Latin America and is classified among the 20 Neglected Tropical Diseases (NTDs),…”; “…The acute phase usually goes unnoticed due to non-specific symptoms and generally lasts 4 to 8 weeks [6]. However, during the chronic phase, which may develop 10 to 30 years after infection,…”; Chagas disease becomes chronic without treatment, and only the drugs nifurtimox and benznidazole, developed more than 50 years ago, are commercially available for its treatment [3]:”
The key for short forms (e.g. Sm-ME, Sm-CP) used in Table 1 must be clearly specified at the end of Table. Also, mention the long forms for Sm-HF1, etc.
Response: The meaning of the abbreviations was included in the legend of Figure 1 (Results section, Lines 87 and 88 of the revised manuscript).
“Figure 1. General scheme for theB bioguided fractionation of S. molle. Data represents IC50 against T. cruzi. Sm-ME: S. molle crude methanolic extract; Sm-HP: S. molle n-hexane partition; Sm-CP: S. molle chloroform partition; Sm-MP: S. molle methanol partition; Sm-HF: S. molle hexane fraction; Sm-CF: S. molle chlorofom fraction.
Can authors comment/explain > 10-fold difference in the IC50 values of Sm-ME and SM-MP? They should have the same content in principle, so this result is confusing.
Response: This issue has been clarified in the Discussion section as follows (lines 158 to 163 of the revised manuscript):
“Ten-fold differences among the IC50 values of Sm-ME and SM-MP were obtained. It may be due to the crude methanol extract (Sm-ME), which contains all non-polar, moderately polar, and polar compounds that were extracted from the plant. Furthermore, in the methanol partition (Sm-MP), only polar compounds of the plant are present, since non-polar and moderately polar compounds were extracted in the hexane and chloroform partitions, respectively. This indicates that the compounds with activity against T. cruzi are mainly non-polar and moderately polar compounds, as confirmed by their identification.”
Can authors report Nifurtimox data in the same table? What was SI for Nifurtimox in this study?
Response: In this study, nifurtimox was not evaluated in Vero cells, and therefore, its Selectivity Index (SI) was not determined.
It is important to include some examples of raw data used to calculate IC50 values in this study. Include one example of control (nifurtimox) and 1-2 examples of the extracts.
Response: We did not consider it necessary to include examples of raw data to calculate IC50 values.
How were the compounds identified after GC-MS? Did authors use any specific database to predict based on m/z? Can authors comment on the relative abundance of the 20 compounds identified?
Response: The method used to identify the compounds has been added to the Materials and Methods section (lines 345 to 350 of the revised manuscript). In addition, m/z values were included in Table 3.
“The GC–MS mass spectrum data were analyzed and identified using the database of the National Institute Standard and Technology (NIST) database to determine their m/z values, according to their 100 % abundance [41]. The chemical classes of the identified metabolites were automatically determined using the Classyfire web-based application (http://classyfire.wishartlab.com, accessed on March 10, 2023) [42].”
The structure of eicosapentaenoic acid needs to be corrected in Figure 2.
Response: Figure 2 was corrected, as suggested (lines 147 and 148 of the revised manuscript).
Cis-5,8,11,14,17-eicosapentaenoic acid

Reviewer 3 Report
Comments and Suggestions for Authors
This study aimed to evaluate the anti-T. cruzi potential of S. molle sub-fractions and identify the possible active compounds. Also, the toxicity of the extracts in various organism was assessed. After reading thorough the manuscript, my impression is heading the toxicity and selectivity index evaluation for further optimisation of the extraction process to get high efficiency with less toxicity ingredient from the plant.
The authors are required to emphasize the main rationale, hypothesis, aim and also the advantage of this study.
Moreover, there are some points which must be clarified before further consideration.
- Figure1 is missing, please check
- Antiprotozoal activity of Schinus molle extract already has been reported, the novelty of this study probably be the assessment of toxicity and the determination of a group of promising active components in the extract.
- In the abstract, Line24-26, I don't think this work identified the exact active compounds included in the extract. Using only qualitative analysis is not sufficient to state "identify". The possible active compounds can be used.
- Line 88, the meaning of SM-HP, SM-CP and SM-MP, SM-ME must be stated.
- The result of hemolysis assay must be represent as the effect of various doses of each extract. Whether the effective dose of the extract which kill the protozoa cause hemolysis induction (no, mild, strong) must be reported. Only IC50 is not suitable to confirm the toxicity in RBC.
- The author must present the rationale why only GC-MS, but not LC-MS was used to identify the candidate active compounds. Moreover, quantitative analysis was not performed. How to make sure whether the compounds found in the extract are major active components or are in the sufficient content to show the activity.
- Discussion must be revised. Each paragraph was separately described without any linkage. The application of knowledge from this study must be explained.
- Due to this study used human RBC, ethic approval is required or not in your country? Or did the author do EXEMPTION?
Comments on the Quality of English LanguageEnglish proofread by native speaker is required.
Author Response
Open Review
(x) I would not like to sign my review report ( ) I would like to sign my review report
Quality of English Language
( ) I am not qualified to assess the quality of English in this paper ( ) English very difficult to understand/incomprehensible ( ) Extensive editing of English language required (x) Moderate editing of English language required ( ) Minor editing of English language required ( ) English language fine. No issues detected
Yes
Can be improved
Must be improved
Not applicable
Does the introduction provide sufficient background and include all relevant references?
( )
(x)
( )
( )
Is the research design appropriate?
(x)
( )
( )
( )
Are the methods adequately described?
(x)
( )
( )
( )
Are the results clearly presented?
( )
( )
(x)
( )
Are the conclusions supported by the results?
( )
( )
(x)
( )
Comments and Suggestions for Authors
This study aimed to evaluate the anti-T. cruzi potential of S. molle sub-fractions and identify the possible active compounds. Also, the toxicity of the extracts in various organism was assessed. After reading thorough the manuscript, my impression is heading the toxicity and selectivity index evaluation for further optimisation of the extraction process to get high efficiency with less toxicity ingredient from the plant.
The authors are required to emphasize the main rationale, hypothesis, aim and also the advantage of this study.
Moreover, there are some points which must be clarified before further consideration.
- Figure1 is missing, please check
Response: Thank you very much for your comment. It was an oversight not to include Figure 1 in the document, which has been included in the revised manuscript.
- Antiprotozoal activity of Schinus molle extract already has been reported, the novelty of this study probably be the assessment of toxicity and the determination of a group of promising active components in the extract.
Response: This issue had been addressed in the Discussion section (lines 152-155 of the revised manuscript):
“In this study, the pharmacological potential of S. molle against T. cruzi was evaluated. Although there are reports of its anti-T. cruzi activity [16,17], only the crude methanol extract has been investigated. However, identification of anti-T. cruzi compounds and their toxicity and SI remained unexploredto be elucidated.”
- In the abstract, Line24-26, I don't think this work identified the exact active compounds included in the extract. Using only qualitative analysis is not sufficient to state "identify". The possible active compounds can be used.
Response: In this study, we did not use qualitative analyses. The phrase “Gas chromatography-mass spectrometry analysis identified 20 compounds” is cited in lines 23 and 24 of the revised manuscript. The compounds present in the plant fraction were identified using GC-MS and database searches. The active compounds are still unknown. However, since reports indicate that these two compounds exhibit antiparasitic activity, it is believed that they may have activity against T. cruzi.
- Line 88, the meaning of SM-HP, SM-CP and SM-MP, SM-ME must be stated.
Response: The meaning of the abbreviations has been included in the legend of Figure 1 (Results section, Lines 85 to 88, as follows:
Figure 1. General scheme for the Bbioguided fractionation of S. molle. Data represents IC50 against T. cruzi. Sm-ME: S. molle crude methanolic extract; Sm-HP: S. molle n-hexane partition; Sm-CP: S. molle chloroform partition; Sm-MP: S. molle methanol partition; Sm-HF: S. molle hexane fraction; Sm-CF: S. molle chlorofom fraction.
- The result of hemolysis assay must be represent as the effect of various doses of each extract. Whether the effective dose of the extract which kill the protozoa cause hemolysis induction (no, mild, strong) must be reported. Only IC50 is not suitable to confirm the toxicity in RBC.
Response: Tables show that hemolytic IC50 values are significantly higher than the IC50 values for T. cruzi, indicating that the concentration required to kill the parasite does not affect erythrocytes.
This issue has been clarified in Table 1 as follows: Line 110: 1This extract, at the IC50 required to kill T. cruzi, did not cause significant hemolysis. 2This extract, at the IC50 required to kill T. cruzi, caused low hemolysis (< 20 %).
And, also, in Table 2 (lines 128-129). 1This extract, at the IC50 required for T. cruzi, did not cause significant hemolysis.
- The author must present the rationale why only GC-MS, but not LC-MS was used to identify the candidate active compounds. Moreover, quantitative analysis was not performed. How to make sure whether the compounds found in the extract are major active components or are in the sufficient content to show the activity.
Response: Regarding the first concern, we decided to use GC-MS instead of LC-MS to identify the compounds in the fraction, due to their nature, as it had been determined that the compounds present were primarily non-polar and moderately polar. About your second issue, we did not include the major active compounds, as explained in lines lines 206 to 209 of the revised manuscript.
Nevertheless,further biotargeted fractionation is required to isolate and identify the bioactive compounds with antiparasitic activity. Moreover, it is essential to evaluate these compounds in an in vivo murine model to validate their antiparasitic efficacy and to exclude any potential toxicity to humans.
- Discussion must be revised. Each paragraph was separately described without any linkage. The application of knowledge from this study must be explained.
Response: Paragraphs have been revised and connected, as suggested.
- Due to this study used human RBC, ethic approval is required or not in your country? Or did the author do EXEMPTION? Response: The ethical statement has now been added to the Materials and Methods section (lines 221 to 227 of the revised manuscript), as suggested:
“4.1. Ethical statement
The procedures used in this study were approved by the Ethics Committee of the Facultad de Ciencias Biológicas (FCB) at the Universidad Autónoma de Nuevo León (UANL), registration no. CI-08-2020. EThe experiments involving human erythrocytes wereas conducted with the informed consent of a healthy donor in compliance with the Official Mexican Technical Standard NOM-253-SSA1-2012. “
Comments on the Quality of English Language
English proofread by native speaker is required.

Round 2
Reviewer 3 Report
Comments and Suggestions for Authors
The author well response to comments and suggestions. This revised version can be accepted for the publication.